# Unveiling the Therapeutic Potential of Kelulut (Stingless Bee) Honey in Alzheimer’s Disease: Findings from a Rat Model Study

**DOI:** 10.3390/antiox13080926

**Published:** 2024-07-30

**Authors:** Ammara Shaikh, Fairus Ahmad, Seong Lin Teoh, Jaya Kumar, Mohamad Fairuz Yahaya

**Affiliations:** 1Department of Anatomy, Faculty of Medicine, Universiti Kebangsaan Malaysia, Cheras, Kuala Lumpur 56000, Malaysia; ammarashaikh9@gmail.com (A.S.); fairusahmad@ukm.edu.my (F.A.); teohseonglin@ukm.edu.my (S.L.T.); 2Department of Physiology, Faculty of Medicine, Universiti Kebangsaan Malaysia, Cheras, Kuala Lumpur 56000, Malaysia; jayakumar@ukm.edu.my

**Keywords:** neurodegenerative disease, stingless bee honey, pot-honey, β-amyloid, neuroprotection

## Abstract

Alzheimer’s disease (AD) poses a major worldwide health challenge because of its profound impact on cognitive abilities and overall well-being. Despite extensive research and numerous clinical trials, therapeutic options remain limited. Our study aimed to investigate the potential of Kelulut honey (KH) as a novel therapeutic agent for addressing the multifactorial pathology of AD. We tried to evaluate the disease-attenuating and neuroprotective potential of KH in the intrahippocampally induced AD rat model by utilizing histochemistry and enzyme-linked immunosorbent assay (ELISA) studies. A total of 26 male Sprague Dawley rats weighing ~280–380 g were randomly divided into three groups: Control, AD-induced (Aβ), and AD-induced and treated with KH (Aβ+KH). The latter two groups underwent stereotaxic surgery, where 6.25 µg of amyloid β_1–42_ peptides were injected intrahippocampally. One-week post-surgery, KH was administered to the treatment group at a dose of 1 g/kg body weight for a period of four weeks, after which the rats went through behavior tests. After completion of behavior analysis, the rats were sacrificed, and the brains were processed for histochemistry and ELISA studies. The open field test analysis demonstrated that KH improved the locomotion of Aβ+KH compared to Aβ (*p* = 0.0013). In comparison, the Morris water maze did not show any nootropic effects on cognition with a paradoxical increase in time spent in the target quadrant by the Aβ group (*p* = 0.029). Histochemical staining showed markedly increased Congo-red-stained amyloid plaques, which were significantly reduced in dentate gyrus of Aβ+KH compared to Aβ (*p* < 0.05). Moreover, significantly higher apoptosis was seen in the Aβ group compared to Aβ+KH (*p* < 0.01) and control groups (*p* < 0.001). Furthermore, the ELISA studies deduced more phosphorylated tau in the diseased group compared to Aβ+KH (*p* = 0.038) and controls (*p* = 0.016). These findings suggest that KH consumption for twenty-eight days has the potential to attenuate the pathological burden of disease while exerting neuroprotective effects in rodent models of AD.

## 1. Introduction

Alzheimer’s disease (AD) is a neurodegenerative condition and the most common cause of dementia, accounting for 60–80% of all cases worldwide [1]. AD is characterized by neurological deposition of amyloid β peptide extracellularly and phosphorylated tau (p-tau) intracellularly [2]. The disease is initiated by increased production of amyloid β peptides and p-tau that lead to oxidative stress and neuroinflammation, both of which further increase amyloid and tau pathology [3]. Once produced, the amyloid β (oligomeric and fibrillar) and p-tau become neurotoxic as they accumulate to form amyloid plaques and neurofibrillary tangles (NFTs), respectively, leading to neurodegeneration by activation of multiple pro-apoptotic pathways [4,5]. Even though AD is widely studied with the pathological pathway and disease progression fairly well understood, it does not have any cure. Moreover, the current management options are single-target therapies that can either improve memory or alleviate amyloid burden [6]. Amid such conditions, traditional medicine has emerged as potent multi-target drug options in the absence of any adverse effects [7]. The disease-ameliorating effect of these nutraceuticals is attributed to the polyphenolic compounds they contain [7]. Honey, being one such nutraceutical, is believed to contain numerous phenolic acids and flavonoids [3,8] and, therefore, exhibits anti-inflammatory, antioxidant, anti-microbial, and healing properties. For this reason, it has long been used for the treatment and management of various illnesses, including gastrointestinal infections, cardiovascular diseases, skin allergies, and wound healing [9,10].

Studies on various types of honey concluded their potential to improve cognitive deficits in humans and rodent models [3]. The comparative studies suggest that Kelulut honey (KH), also known as stingless bee honey, has higher phenolic compound and flavonoid content and stronger antioxidant capacity compared with other kinds of Malaysian honey [11,12,13]. KH is produced by two genera of stingless bees, Melipolini and Trigona [14,15]. It has shown promise to exert antioxidant, anti-inflammatory, and anti-apoptotic effects in the central nervous system (CNS), attributed to its high polyphenolic content [16,17]. Due to the mentioned attributes, KH is found to be a potent nootropic and neuroprotective agent [16,18].

To our knowledge, only two studies discuss the effects of KH on CNS in rodents [16,18]. These studies deduced KH’s capability to prevent neuronal damage and improve cognition in healthy mice [18] and metabolic syndrome (MetS)-induced rats [16]. These studies provide important insights, but the impact of honey on CNS diseases like Parkinson’s disease and AD remains an open question. Moreover, current therapies in clinical practice focus on symptom management rather than addressing the underlying causes. Exploring alternative approaches, such as those offered by traditional medicines, could prove valuable in the search for both preventive and therapeutic interventions.

Therefore, this research tested the effect of KH on AD-induced rat models. For this purpose, an AD model was reproduced by following an intrahippocampal approach to administer amyloid β_1–42_ [19,20]. Subsequently, the abnormalities in behavior, burden of pathological markers, and hippocampal neuronal count were compared in various groups to analyze the potential of KH to ameliorate pathological damage in neurodegenerative disorders.

## 2. Materials and Methods

### 2.1. Animal Grouping

This study used a total of 26 male Sprague Dawley rats around 3 months of age (weighing ~280–380 g) supplied by the Laboratory Animal Research Unit, Universiti Kebangsaan Malaysia (UKM). The rats were randomly divided into three groups: Control (*n* = 9), AD-induced (Aβ) (*n* = 8), AD-induced and treated with Kelulut honey (Aβ+KH) (*n* = 9). Throughout the study, the rats were kept in the Animal Unit, Department of Anatomy, Faculty of Medicine, UKM, in a conventional air-conditioned laboratory environment at a temperature of 25 to 27 °C and on a 12 h day–night cycle. Each rat was placed in a separate cage and was provided with rat pellets and tap water ad libitum for 14 days as an acclimatization period to adapt to the new place before the commencement of the study. All procedures were carried out in accordance with the institutional guidelines for animal research surgical procedures of Universiti Kebangsaan Malaysia Research and Animal Ethics Committee (UKMAEC) with approval number: ANAT/2022/FAIRUZ/20-JULY/1255-JULY2022-AUGUST-2023. The study protocol was based on studies from Yuan et al. (2016) and Forero et al. (2023) with some modifications [20,21].

### 2.2. Amyloid β_1–42_ Peptide Incubation

The whole volume of 1 mg human amyloid β_1–42_ provided in an ampule (Targetmol^®^, Boston, MA, USA) was dissolved in 20 μL of dimethyl sulfoxide (Heiltropfen) to make ~21 μL of the final solution. The master mix solution was divided into 2.5 μL aliquots. Seven of them were kept at −80 °C, while one was diluted at a ratio of 1:19 in distilled water (dH_2_O), which was 2.5 μL into 47.5 μL of dH_2_O, that formed 50 μL of the working solution. Each 1 mL microcentrifuge tube (Eppendorf, Hamburg, Germany) contained 125 μg of amyloid β_1–42_ working solution wrapped in aluminum foil and incubated at 37 °C for 7 days. A dose of 6.25 μg of amyloid peptides was chosen to be injected into each hippocampus which was modified from a previous study by Facchinetti et al., the dose was increased to ensure the development of AD [19].

### 2.3. Stereotaxic Surgery

The rats from Aβ and Aβ+KH groups were weighed before they underwent stereotaxic surgery. The rats were anesthetized using an intraperitoneal injection of ketamine–xylazine diluted in dH_2_O (1:4 ratio); the dosage was selected to be 0.5 mL/300 g body weight. The anesthetized rat was fixed to the stereotaxic apparatus after its head was shaved. The shaved area was sterilized using 75% ethanol. A subcutaneous injection of 0.2 mL lidocaine was given in the area of surgery, and an eye cream was applied to prevent corneal drying, after which a longitudinal incision was made to expose the cranium.

The dorsal hippocampus was localized on both sides in relation to bregma according to the Paxinos and Watson rat brain atlas at AP = −3.8 mm and ML = ±2.2 mm. A burr hole was drilled on both sides. Each hippocampus was injected at DV = −3.4 mm with a volume of 2.5 µL incubated solution containing 2.5 µg of amyloid β_1–42_/µL using a 10 µL microinjector (Hamilton^®^, Reno, NV, USA) at a speed of 0.5 μL/min. The microinjector was held at the site for an additional 2–3 min to prevent reflux. Next, the microinjector was removed, after which the wound was stitched with a self-absorbable suture, and an antiseptic cream was applied. In addition, 0.5 mL of 0.9% normal saline was injected subcutaneously to prevent dehydration. The rat was placed on a heating pad to maintain body temperature until it gained consciousness and was later shifted to its cage.

### 2.4. Kelulut Honey Treatment

After one week of recuperation time post-surgery, the Aβ+KH group was force-fed KH at a dose of 1 g/kg body weight of KH daily for 4 weeks (28 days). The selected dose was previously optimized in our study which deduced the neuroprotective effects of KH on MetS-induced rats [16]. The KH was diluted in dH_2_O (1:1 ratio) and was administered by using an 18 G feeding needle. The needle was kept inside for 5 s after forced ingestion of KH to prevent reflux. Each rat was observed for 15 min after oral gavage to ensure that no harm was done during the procedure. The body weight of the Aβ+KH group rats was recorded every alternate day to ensure that the amount of KH given was proportional to the change in weight. Once the 28-day treatment period was completed, the behavior tests were conducted. The behavioral assessment was commenced 24 h after the last dose of KH.

### 2.5. Behavior Assessment

#### 2.5.1. Open Field Test

The open field test (OFT) was utilized to measure locomotor activity and anxiety. For the test, a plexiglass black-colored box measuring 40 cm × 40 cm × 30 cm was used. This experiment was conducted in a laboratory having a controlled environment with a temperature of 24 ± 1 °C. The corner area of the room was illuminated by fluorescent lights with illumination conditions of 30 lux to reduce stress on the rats and to prevent the reflection of light from interfering with both motion detectors and the recording of the experiments. Before starting the experiment, the study rats were transferred to the experiment place and allowed to habituate for 1 h. General activity levels were observed by measuring horizontal motor activity (movement across the field), and the total distance traveled during 5 min of exposure to the arena. The videos were analyzed using SMART software (version 3.0.06, Panlab, Holliston, MA, USA). Anxiety was assessed by thigmotactic behavior (i.e., more time spent in the peripheral zone compared to the central zone).

#### 2.5.2. Morris Water Maze Test

Twenty-four hours after the one-day OFT, the Morris water maze (MWM) test was conducted. The MWM is a method used to study spatial learning and memory, both of which depend on the normal functioning of the hippocampus and associated pathways [22,23]. The experiments were conducted using a black domestic cylindrical water tank measuring 121 cm in diameter and 35 cm high. In addition, a black platform measuring 12 cm × 12 cm × 27 cm was used. The tank was placed in a sound-proof room with fluorescent lights with the same illumination levels as the OFT and placed in the corners to avoid reflections on the water surface. The room had a controlled environment with a temperature of 24 ± 1 °C and had clear indications such as exhaust fans and wall posters. The tank was filled with tap water at room temperature until the platform was submerged 1.5–2 cm below the water surface.

The platform was placed in the middle of one of the quadrants, with its position being constant throughout training day sessions. Each rat was released into the tank at a variable starting position. Four release locations, which were equal distances from the platform, were randomly selected for each trial of training days. The experiment comprised five training days with each day having four cycles (with a maximum of 60 s given to find the platform in each cycle), followed by a probe test day (24 h after the last training day). On the probe test day, all rats were released from the same location and their movement in water was recorded for 60 s without a platform. A motion detector and video recorder were installed on top of the tank at a height of 1.5 m to record all data. The recorded videos were then analyzed using the SMART software (version 3.0.06, Panlab, Holliston, MA, USA).

### 2.6. Brain Sectioning and Storage for Assay

After completion of the behavior tests, the rats were sacrificed to obtain brains for analysis. One of the two cerebral hemispheres was incised to collect the hippocampus which was subsequently stored in a labeled Eppendorf tube. The tube was placed on dry ice and sprayed with 70% ethanol to snap-freeze the sample. The frozen samples were kept at −80 °C until processed for enzyme-linked immunosorbent assay (ELISA). Before being homogenized, the samples were thawed on ice, and the residual blood was removed by washing the tissue with pre-cooled phosphate-buffered saline (PBS). Next, the samples were weighed and homogenized with PBS. A volume of 4.5 mL PBS was added to the 0.5 g of hippocampal tissue with the volume adjusted according to varying weight. To prepare homogenate, the tissue within the PBS was manually homogenized using an iron bead homogenizer. Prepared homogenates were centrifuged at 5000× *g* for 5 min at 4 °C in a refrigerated centrifuge. The supernatant was immediately collected using disposable dropping pipettes and stored at −80 °C in labeled Eppendorf tubes until the day of the ELISA reading. A rat Aβ_42_ ELISA kit (Fine test, Wuhan, China, Catalogue no. ER0755), rat p-tau ELISA kit (Fine test, Catalogue no. ER1304), rat NF-κB p65 ELISA kit (Fine test, Catalogue no. ER1187), rat superoxide dismutase 1 (SOD1) ELISA kit (Fine test, Catalogue no. ER0332), and rat malondialdehyde (MDA) ELISA kit (Fine test, Catalogue no. ER1878) were used in this study.

### 2.7. Brain Fixation and Embedding in Paraffin Section

For histochemistry, the other hemisphere was cut on the coronal plane to separate the cerebellum. The cerebrum (area of interest) was stored in a labeled container with 10% neutral buffered formalin for fixation to be used later. To make the paraffin sections, the formalin-fixed brain tissue was processed using an automatic vacuum tissue processor machine (Leica, Singapore, model no. STP120). Briefly, the steps of the tissue processing included dehydration (by sequential immersion in alcohol), clearing (by immersion in toluene), and wax infiltration (by paraffin immersion). Once the processing cycle was completed, the tissue was embedded in paraffin wax to form blocks.

### 2.8. Hematoxylin and Eosin

The tissue sections of 5 µm thickness were used for hematoxylin and eosin (H&E) staining. H&E staining was performed by using the autostainer XL machine (Leica, model no. ST5010). The steps included deparaffinization by two changes in xylene which were followed by sequential rehydration of the tissue in alcohol, and then hydration with tap water. The slides were then stained with the hematoxylin dye and were subsequently washed in tap water. Slides were dipped in acid alcohol and washed again with tap water. Next, the slides were immersed in dH_2_O followed by eosin dye after which the slides went through sequential dehydration. After completion of the H&E staining, the slides were manually mounted and left to dry. H&E-stained slides were observed under an HD light microscope (Carl Zeiss Primo star, Oberkochen, Germany) to look for the injection site.

### 2.9. Congo Red Stain

Congo red staining was performed using the Congo red stain kit (Abcam, Waltham, MA, USA, Catalogue no. ab150663). The tissue sections of 6 μm thickness were chosen for staining after optimization from the AD human brain. For staining, the slides were deparaffinized by immersion in two changes of xylene followed by serial rehydration through alcohol and then in dH_2_O. Next, 4–6 drops of hematoxylin were added to the slides, and then the slides were incubated for 5 min at room temperature (RT). After rinsing in tap water, the bluing agent was added and slides were incubated for 30 s at RT. Slides were then washed in dH_2_O, subsequently dipped in 95% alcohol, and applied with Congo red solution for 20 min at RT. Last, the slides were washed with 100% alcohol followed by xylene, and then mounted to be visualized under the HD light microscope.

### 2.10. TUNEL Assay

To observe the apoptotic cells, the slides with 5 μm thick sections were stained by using a one-step terminal deoxynucleotidyl transferase (TdT) dUTP nick-end labeling (TUNEL) in situ apoptotic kit ((green, FITC), Catalogue no. E-CK-A320, Elabscience, Houston, Texas, USA). The slides were stained according to the manufacturer’s protocol. Briefly, first, the sections were deparaffinized and hydrated in PBS. Next, 100 μL of 1x proteinase K working solution was added to each slide and incubated at 37 °C for 20 min after which the slides were washed with PBS.

For labeling, 100 μL of TdT equilibration buffer was added to each sample, and the slides were incubated at 37 °C for 20 min. The labeling working solution was prepared just before use and 50 μL was added to each slide. The slides were incubated at 37 °C for 60 min in a black box. After washing with PBS, slides were stained with 20 μL of 4′,6-diamidino-2-phenylindole (DAPI) working solution and left for 5 min at RT. The slides were again washed with PBS and were mounted when dried. The slides were visualized under a fluorescence microscope using a standard fluorescence filter to observe green fluorescence at 520 ± 20 nm and blue fluorescence at 460 nm for DAPI.

### 2.11. Image Analysis

All pictures were analyzed using the ImageJ software Version 1.50i (https://imagej.net/ij/; accessed on 1 November 2023). Plaque burden and apoptotic cell count were observed by Congo red and TUNEL assay staining, respectively. Hippocampal areas cornu ammonis (CA)1, CA3, and dentate gyrus (DG) were selected as the regions of interest. The apoptotic neurons were counted in the pyramidal layer of CA1 and CA3, while the plaques were counted in the whole region of CA1, CA3, and DG.

### 2.12. Statistical Analysis

The test result data were calculated as mean ± standard error of the mean (SEM). Statistical Package for Social Sciences version 29.0 (SPSS) was used to perform statistical analysis. The Kolmogorov–Smirnov test was utilized to determine the data distribution curve. The data were analyzed using one-way ANOVA, however, the Kruskal–Wallis test was used to analyze Congo red staining results due to non-normally distributed data. A value of *p* < 0.05 was considered significant.

## 3. Results

### 3.1. Open Field Test

Path length was the distance covered within the arena and represented the locomotor activity of the study rats. The Aβ group had a significantly shorter path length compared to controls (*p* = 0.0013). Moreover, the path length in the Aβ+KH group was statistically insignificant compared to the control group (*p* = 0.281). Next, the thigmotaxic behavior was analyzed. The amount of time spent in the central zone is inversely proportional to anxiety, which means the more time a rat spent in the central zone, the less anxious it was. The results of the OFT showed that the time spent in the peripheral zone was similar among the three groups. In comparison, less time was spent in the central zone by the Aβ group compared to the control, however, the difference was not statistically significant (*p* = 0.336) (Figure 1).

### 3.2. Morris Water Maze Test

There was no difference in the total distance and escape latency during the 5-day training period. Moreover, the swim speed did not differ significantly between groups on any day. Similarly, the probe test showed no significant difference in total distance covered and platform area crossings. However, both the AD-induced groups spent more time in the target quadrant. The difference was statistically significant between Aβ and the control group (*p* = 0.029) (Figure 2).

### 3.3. H&E and Congo Red Staining

The 5 µm brain sections were stained with H&E to identify the injection site within the brain. The injection site was observed to be located in area CA1 and appeared as scarring at the site of injection with inflammatory cell infiltration at the place of inoculation.

Next, the 6 µm thick sections were selected for Congo red staining after optimization using the AD human brain (Figure 3). The experimental rat brain analysis showed no staining in the control samples, whereas a significant amount of staining was observed in the Aβ and Aβ+KH groups (Figure 4). On analysis, a high number of amyloid plaques was observed in Aβ with a decreasing trend in all areas of interest, however, only DG showed a significant reduction in Aβ+KH after KH treatment (*p* < 0.05) (Figure 4).

### 3.4. TUNEL Assay

The TUNEL-assay-positive cells appeared bright green in color and were confirmed as turquoise-green cells after merging with the DAPI stain (Figure 5). The total number of TUNEL-positive cells in the Aβ group was significantly high compared to the control (*p* < 0.001) and Aβ+KH group (*p* < 0.01). The difference in TUNEL-positive cells in the areas of interest: CA1, CA3, and DG was also analyzed. In area CA1, the Aβ group had significantly more TUNEL-positive cells than both the Aβ+KH (*p* < 0.05) and control group (*p* < 0.01). Similarly, in area CA3, the increase in apoptotic cell count was significant between the Aβ group in comparison to both the Aβ+KH (*p* < 0.05) and control group (*p* < 0.001). Likewise, in DG, there were more TUNEL-positive cells compared to the Aβ+KH group (*p* < 0.05) and control group (*p* < 0.001). In contrast, the apoptotic count in CA1, CA3, and DG was not significant in Aβ+KH compared to the controls (*p* > 0.05).

### 3.5. ELISA Results

The ELISA studies were performed using the homogenate of the rat hippocampus. The parameters included were two disease markers: amyloid β_42_ and p-tau, two oxidative stress markers: SOD1 and MDA, and a marker of neuroinflammation: NF-κB p65.

The ELISA results of p-tau showed a significantly increased concentration in the Aβ group compared to the control group (*p* = 0.016). The p-tau levels in Aβ were also significantly higher than in the Aβ+KH group (*p* = 0.038). Moreover, the KH supplementation appeared to have a preventive effect on p-tau as the levels in Aβ+KH were comparable to the controls (*p* = 0.863). Other ELISA tests (amyloid β_42_, SOD1, MDA, and NF-κB p65), however, did not show any difference between groups (Figure 6).

## 4. Discussion

This study is the continuation of two studies titled ‘Effect of Kelulut Honey on the Hippocampus of MetS-induced Rats’ [16] and ‘Comparable Benefits of Stingless Bee Honey and Caffeic Acid in Mitigating the Negative Effects of MetS on the Brain’ [17]. Therefore, this study utilizes the same dose of KH that was optimized previously (1 g/kg) to attenuate the neuronal damage in MetS-induced rats [16]. We expected to see the AD-attenuating potential of KH attributed to its polyphenolic content, as we reviewed in our previous paper [3]. So, this study aimed to compare the ability of KH as an anti-amyloid, anti-tau, antioxidative, anti-inflammatory, and neuroprotective agent in AD-induced rat models.

### 4.1. Use of Male Sprague Dawley Rats and Amyloid β_1–42_ Peptides for the Study

This study recruited rats aged around three months (young adult phase) and weighing ~280–380 g. Male Sprague Dawley rats were used to avoid the influence of gender, specifically the estrogen hormone that is preventive against amyloid plaque formation and AD initiation [24,25]. The choice of amyloid β_1–42_ over amyloid β_1–40_ and other variants was based on its longer length and, therefore, higher tendency to aggregate and increased neurotoxicity potential compared to all other fragments [26,27]. Similar amyloid β peptide (i.e., amyloid β_1–42_), incubation time, and dose for the establishment of AD were used by several studies that successfully induced AD in rodent models [19,20,26]. The Congo-red-positive staining was compared with the studies that readily stained amyloid plaques in transgenic mice and AD-induced rat brains [28,29]. Similarly, the findings of TUNEL-assay-positive cells were found to be in line with the studies that found increased TUNEL-assay-positive apoptosis in transgenic mice [30] and AD-induced rats [31].

### 4.2. Effect of Kelulut Honey on Behavior Domains

The locomotion in the OFT shows the exploratory nature of research animals and is considered to be a marker of normal excitatory behavior. Previous studies have found both decreased [32] and increased locomotor activity [33,34] in transgenic AD mice. Researchers in all the mentioned studies deduced the change in locomotor activity (high or low compared to controls) as a marker of anxiety. The results of our experiment also deduced significantly reduced locomotor activity (represented by the path length) in the Aβ group during the 5 min OFT (*p* < 0.01).

In addition, the percentage of time spent in the central zone or the percentage of time spent investigating the central area of the open field box also gives an indication of the anxiety level of the study animal [35]. Animals that are anxious or afraid spend less time in the middle of the field and more time next to the wall, reducing the ratio of time in the center compared to the total time or ‘center to total time ratio’. Our study showed less time spent in the central zone by the Aβ group, however, the results of the latter were not significant. Moreover, in terms of the development of AD, the decreased exploration of the central zone and reduced locomotion are possibly indicative of the initiation of AD, as mentioned in the studies on 3xTg-AD transgenic mice, where the mice had significantly less time spent in the central zone with markedly reduced locomotion [32,36].

Next, learning and memory, which are components of cognition, and depend on the normal functioning of the hippocampus were tested. The MWM test was utilized to observe the deficits of cognition as observed in AD. Our experiment results showed no difference in the total distance covered, escape latency, and swim speed between the groups during the 5-day training period. Generally, all three groups were similar in the distance covered and the escape latency; the rats (from all groups) took the most time and the longest path on the first day, while markedly reducing both the time and distance from the second day onwards. However, no such trend was observed in the results of the swim speed. These findings are in line with the results of previous studies that did not find any difference in the escape latency of 2–3-month-old transgenic mice during the five-day training period. Similarly, the results of the total distance covered and swim speed are supported by previous studies that did not find any cognitive deficits even in 2–3-month-old transgenic models [37,38,39].

Moreover, transgenic AD rodents are reported to cover more distance while traveling to the platform area in probe test [38,40]. During the probe test of our experiment, the total distance covered and the number of times the platform area was crossed did not differ between groups. However, the time spent in the target quadrant was surprisingly higher for the AD-induced groups, where the Aβ group showed significantly increased time compared to the control (*p* < 0.05). These results can be supported by similar studies where the young transgenic and AD-induced rodents spent more time in the target quadrant than the other quadrants [20,37,38]. As the time spent in the target quadrant is directly correlated with the cognition status, the more time spent by the AD rodents is suggested to be due to the cognition-maintaining effect of increased amyloid β_42_ [41] or compensatory increased connectivity between the hippocampus and prefrontal cortex in the early stages of AD [42].

Taken together, since the significant difference in OFT path length was not supplemented by any significant results of thigmotaxic behavior, the anxiolytic potential of KH was not certain. In addition, our study could not evaluate KH’s potential for memory improvement from the results of the MWM test.

### 4.3. Effect of Kelulut Honey on Amyloid Plaque Burden

Congo red is highly specific for amyloid β aggregates. Therefore, despite having lower sensitivity, it is the standard agent used to identify amyloid deposition in tissues [43]. Several studies have used Congo red to stain the AD-induced brain [44,45,46], where the staining results were similar to those of our staining of the post-mortem AD human brain [47].

Similarly, the Congo red staining of our experimental rat brain tissue displayed amyloid plaques developed in rats that underwent amyloid β induction; the amyloid staining was confirmed by comparing the tissue with the Congo-red-stained human AD brain sections. On analysis, we found that the 6-week post-surgery period was sufficient to significantly increase the number of amyloid plaques in the Aβ and Aβ+KH groups when compared with the control group. Furthermore, the analysis of the Aβ+KH group showed that the KH treatment for 28 days can markedly reduce the amyloid plaque burden that was significantly low in DG of the Aβ+KH compared to the Aβ group (*p* < 0.05) (Figure 4).

Although the results are suggestive of the KH potential in mitigating plaque burden, it can be due to attenuation, prevention, or a combination of both. That said, since the KH treatment was started just after a week of amyloid β induction and lasted for four subsequent weeks, the probable mechanism is suggested to be more of a preventive type as proposed in studies deducing preventive effects of honey on amyloid-β-induced neurotoxicity [48] and dementia [49].

### 4.4. Results of TUNEL Assay Analysis

The purpose of the TUNEL assay study was to observe the hippocampal neuron apoptosis and assess the neuroprotective potential of KH. The hippocampus is an area of the brain where learning occurs and memories form. The hippocampus, especially area CA1, is the most sensitive region affected by various types of dementia including AD [50]. Several studies in the last couple of decades have deduced the ability of polyphenols to cross the blood–brain barrier (BBB) and act as neuroprotective agents [51,52,53]. Although KH cannot cross the BBB, it can exert its antioxidant, anti-inflammatory, and neuroprotective effects on the brain as deduced in our previous studies [16,17].

The results of our study show that KH possesses neuroprotective potential, as the Aβ group had significantly increased apoptosis in all three regions of interest compared to both Aβ+KH and control groups. In addition, the apoptotic cell counts in the Aβ+KH group were statistically insignificant compared to the control group. Our study deduced that KH consumption at a dose of 1 g/kg body weight for 28 days exerts neuroprotective effects in the hippocampus by preventing apoptosis that follows amyloid plaque and NFT deposition. Since the cells in the areas CA1, CA3, and DG, along with the subiculum, are important in memory encoding and retrieval [54], long-term use of the KH may result in better cognitive performance in AD subjects.

### 4.5. ELISA Test Results

Amyloid β and p-tau are the characteristic pathological markers of AD. We expected significantly higher levels of amyloid β_42_ and p-tau in the AD-induced groups and very low or undetectable levels in the control group. However, surprisingly, the control group showed substantial levels of amyloid β_42_ and p-tau. Although unexpected, these results of the control can be supported by the previous research which deduced that there is a production of different variants of soluble amyloid β (like amyloid β_42_, amyloid β_40_, etc.) in the healthy brain throughout life [55,56], but it is their overproduction and subsequent impaired clearance that lead to deposition [57] and development of AD. Similar to amyloid β_42_, there is the formation of p-tau, while the total tau is maintained below the baseline level in healthy brains; however, increased production of p-tau leads to high levels of total tau and accumulation in the AD brain [58]. Additionally, tau is phosphorylated in as many as 45 different sites in the AD brain, whereas p-tau present in healthy brains is phosphorylated in ~10 sites [59].

The comparative analysis between the three groups showed that, despite no significant difference, there was a trend of increased amyloid β_42_ levels in the hippocampus of the Aβ group compared to the Aβ+KH group, whereas the levels in the latter were similar to those of the control group.

Unlike the results of amyloid β_42_, the p-tau levels were found to be significantly increased in the Aβ group compared to both Aβ+KH and control groups, while the comparison of the latter two groups showed nearly similar levels. These results point towards a possible involvement of two (or more) pathways in AD pathogenesis. This study suggests that the tau pathology precedes the amyloid pathology or is an independent phenomenon—a hypothesis supported by various newer studies [60,61,62].

Our study indicates that KH is potent enough to mitigate p-tau levels in the AD hippocampus and likely in the AD-induced rat brain when consumed daily for four weeks. In comparison, the findings of the amyloid β_42_ suggest that although the levels of soluble amyloid β_42_ are not significantly different between the three groups, the assembly of amyloid fibrils to form plaques is significantly reduced in DG, along with a marked reduction in other areas, after KH consumption. By considering the results of the Congo red staining to assess amyloid burden, it seems that the KH likely attenuates plaque deposition before (and to a greater extent than) exerting effects on the total soluble amyloid levels. Moreover, our study also points towards the likelihood that the amyloid deposition begins as soon as there is more production of amyloid β, even before the soluble amyloid β levels are significantly increased from the baseline levels.

As SOD1 is an antioxidant, its levels are increased in conditions promoting oxidative stress, however, the levels are decreased in AD due to an impaired antioxidant capacity [63,64,65]. Most studies found a decrease in SOD1 levels in various regions of the AD brain, including the temporal and frontal lobes, and the hippocampus [63,64,65]. The results of this study are in line with the results of many prior studies that deduce a reduction in SOD1 levels in AD [63,64,65]. Analyzed results suggest that, although not significant, the induction of AD caused a decrease in SOD1 in the Aβ group compared to the controls. Moreover, the KH intake increased SOD1 in the Aβ+KH group above the control levels. The tendency to increase SOD1 after KH consumption seems to be similar to the deduced results of other polyphenolic compounds, e.g., resveratrol [66]. Again, possibly due to the shorter duration of the study, none of the results is statistically significant.

MDA is considered an important marker of oxidative stress in neurodegenerative diseases [67,68]. AD is characterized by a poor oxidative-stress defense mechanism, indicated by a higher level of MDA [66,69]. As a product of lipid peroxidation, MDA is also found in healthy brains [70,71,72]. Since all three groups in our study were nearly identical with regard to MDA levels, the results suggest that increased lipid peroxidation and MDA production possibly are late events in the AD pathogenesis.

NF-κB p65 is an important amplifier of AD pathogenesis that exerts its effects by binding to the κB elements on the promoter of beta-site amyloid precursor protein (APP) cleaving enzyme 1 (BACE1) to increase the expression of β-secretase [73]. The induction of BACE1 expression, subsequently, shifts APP processing to the amyloidogenic pathway and leads to the increased formation of amyloid fibrils [74]. Additionally, NF-κB contributes to tau pathology [75] and neuronal apoptosis [76]. By the mentioned mechanism, NF-κB, especially the factor p65, augments AD pathogenesis by inducing multiple targets.

NF-κB is present in the healthy brain and plays a fundamental role in various physiological cellular functions like regulating cell cycle, proliferation, and cell death [77,78]. Therefore, its baseline levels were expected in control brains. However, unexpectedly, this study found high levels of NF-κB p65 in all three groups, including the control group. While the Aβ group had almost similar levels of NF-κB p65 compared to controls, a slight trend of decrease in the inflammatory marker levels was observed after KH ingestion. A study comprising a longer duration of KH consumption and simultaneous AD development would likely give well-demarcated results and a clearer picture of the neuroinflammation in animal brains.

Based on our results, KH seems to be a better substitute for the available Food and Drug Administration (FDA)-approved therapeutic choices that are specifically targeted to only one aspect of the disease pathology (Figure 7). As KH can exert effects on various aspects of disease pathophysiology in addition to ameliorating cognitive deficits and anxiety [16], it has the potential to be a more effective treatment option.

## 5. Conclusions

This study concludes that KH can inhibit the AD pathology affecting the hippocampus by exerting neuroprotection, attenuating amyloid plaque deposition, and reducing p-tau levels. These effects of KH are likely attributed to its high polyphenolic content. These nutraceutical effects precede KH’s ability to mitigate oxidative stress and neuroinflammation. Along the same line, the neuroprotective and disease-attenuating effects of KH would possibly be visible in behavior tests after a longer duration of consumption.

The likely mechanism of action of KH suggested from this study is neuroprotection resulting from attenuation of the amyloid and tau pathology (Figure 8). Our study deduced that consumption of KH for 28 days can decrease tau hyperphosphorylation and NFT formation while decreasing aggregation of fibrillar and oligomeric amyloid β into plaques. The neuroprotective effects are possibly the result of restriction of the pro-apoptotic pathways that are directly initiated by the plaques and NFT-induced neurotoxicity.

The study had a few limitations, including an insufficient duration to adequately examine oxidative stress, neuroinflammatory damage, and behavioral changes. The lack of CSF analysis further restricted insights into oxidative stress and neuroinflammatory markers. Extending the study duration would allow for a more in-depth investigation of these factors.

## Figures and Tables

**Figure 1 antioxidants-13-00926-f001:**
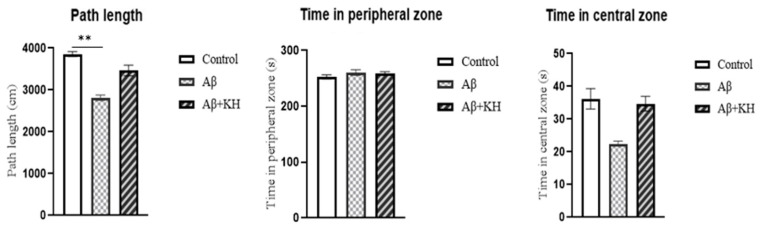
Path length and time spent in peripheral vs. central zone in OFT arena. ** *p* < 0.01.

**Figure 2 antioxidants-13-00926-f002:**
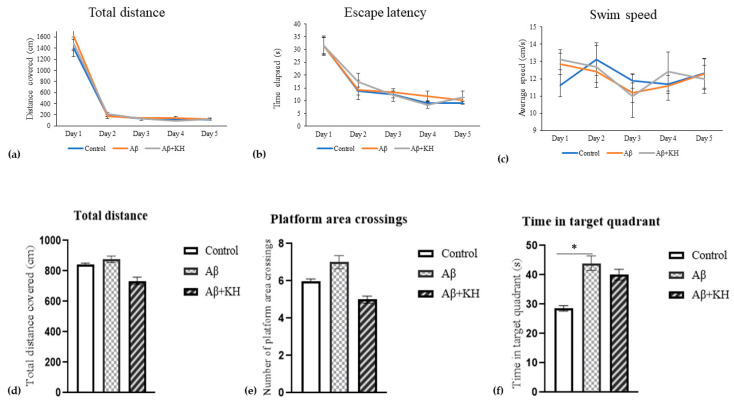
Results of the training days. (**a**) Total distance (**b**), escape latency, (**c**) swim speed, and probe test (**d**) total distance covered, (**e**) number of times the platform area was crossed, and (**f**) the time spent in target quadrant. * *p* < 0.05.

**Figure 3 antioxidants-13-00926-f003:**
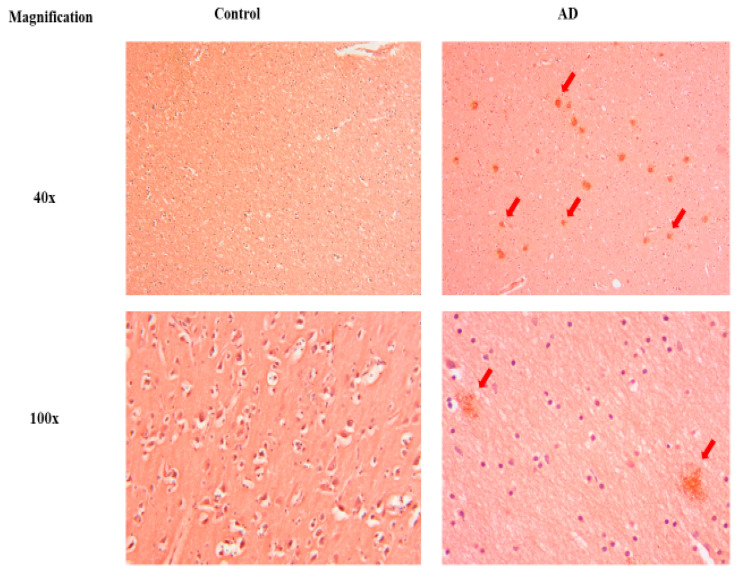
Congo-red-stained human brain with (AD brain) and without AD (control brain). The orange-red stained amyloid plaques can be seen in 40× and 100× magnification (red arrows).

**Figure 4 antioxidants-13-00926-f004:**
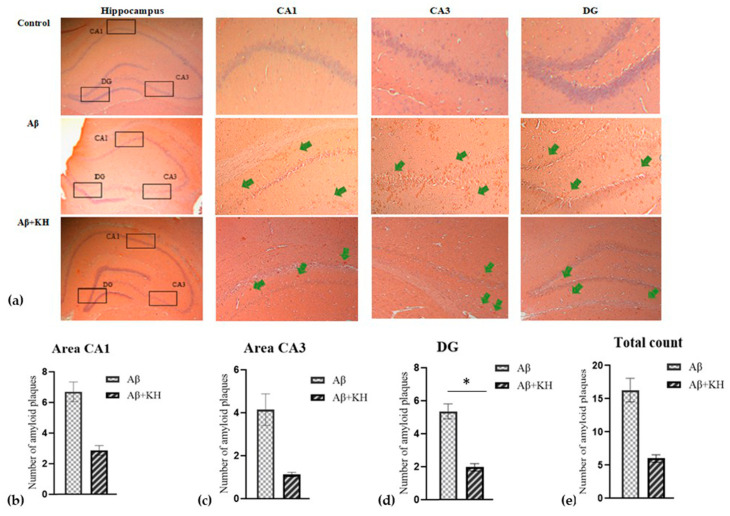
(**a**) Congo-red-stained control, Aβ, Aβ+KH groups (magnification 40×). The areas CA1, CA3, and DG are further enlarged (magnification 100×) to show the amyloid plaques (green arrows). Congo-red-stained amyloid plaque count in areas (**b**) CA1, (**c**) CA3, (**d**) DG and (**e**) the total count in all regions combined. * *p* < 0.05.

**Figure 5 antioxidants-13-00926-f005:**
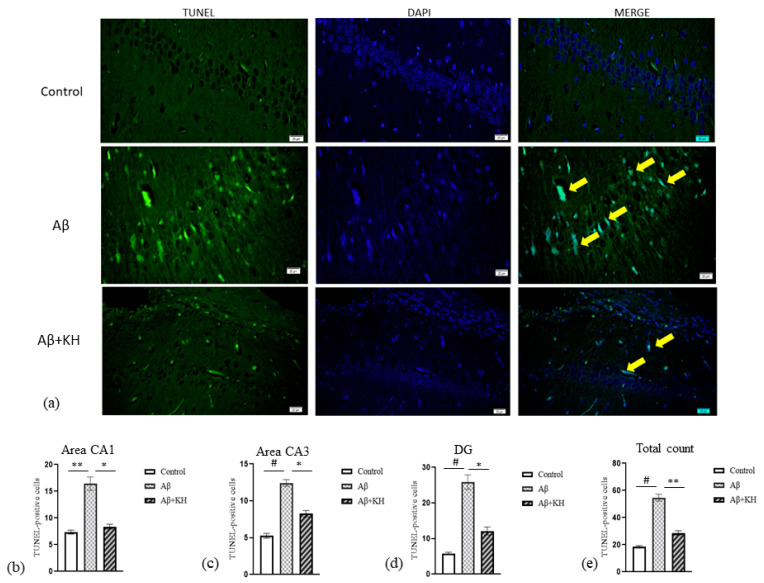
(**a**) TUNEL assay of area CA1 of control, Aβ, Aβ+KH groups (magnification 100×). Bright green TUNEL-positive cells appear turquoise green after merging with DAPI (yellow arrows). Number of TUNEL-positive cells in areas (**b**) CA1, (**c**) CA3, (**d**) DG, and (**e**) the total count in all three regions. * *p* < 0.05, ** *p* < 0.01, # *p* < 0.001.

**Figure 6 antioxidants-13-00926-f006:**
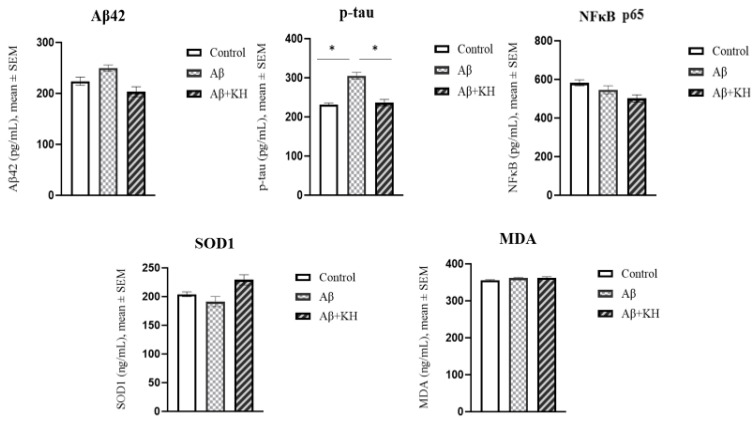
The ELISA results of Aβ42 (amyloid β_42_), p-tau, NF-κB p65, SOD1, and MDA in rat hippocampus. * *p* < 0.05.

**Figure 7 antioxidants-13-00926-f007:**
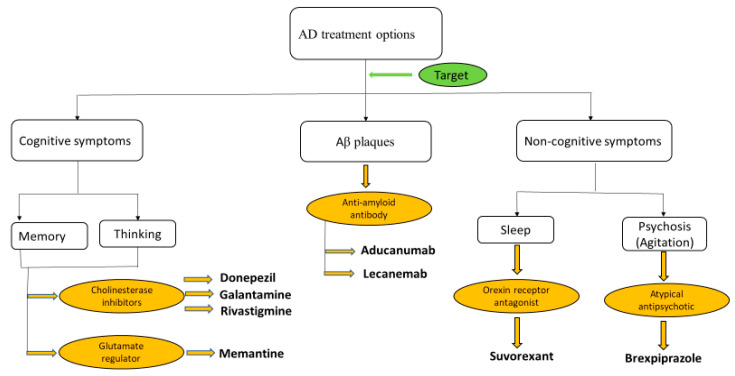
Current FDA-approved treatment options targeting pathology and symptoms of AD.

**Figure 8 antioxidants-13-00926-f008:**
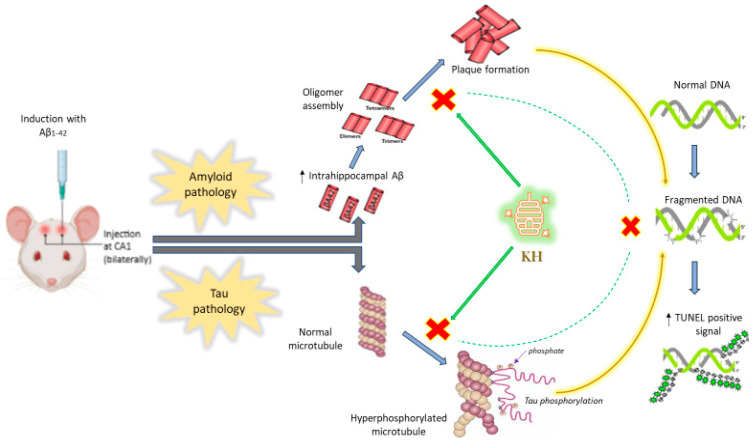
Possible mechanism of action of KH to ameliorate AD burden in amyloid β_1–42_ (Aβ_1–42_)-induced models. Figure showing Aβ_1–42_ intrahippocampal injection initiating amyloid and tau pathologies with subsequent formation of amyloid plaques and p-tau (leading to microtubule dysfunction); both the pathological markers later cause DNA damage (evident as increased TUNEL-positive signals). The KH treatment likely targets plaque formation and tau hyperphosphorylation (shown with the red cross) that ultimately results in decreased neuronal apoptosis (shown with dotted green line).

## Data Availability

The raw data supporting the conclusions of this article will be made available by the authors on request.

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
