# Peer review of "Unveiling the Therapeutic Potential of Kelulut (Stingless Bee) Honey in Alzheimer’s Disease: Findings from a Rat Model Study"

_antioxidants, 2024, doi:10.3390/antiox13080926_

Round 1

Reviewer 1 Report

In this study, the authors reported, “Unveiling the therapeutic potential of Kelulut (stingless bee) honey in Alzheimer's disease: findings from a rat model study.”.

Some critical comments and suggestions

1. The abbreviation should be explained before it is abbreviated. For example, MWM is mentioned in the abstract?

2. Further details regarding the biomedical characteristics of kelulut honey are needed in the introduction.

3. On what basis was the dose fixed? Why only one particular concentration was chosen?

4. Why not compare it with standard drugs?

Some critical comments and suggestions

1. The abbreviation should be explained before being abbreviated. For example, MWM is MWM mentioned in the abstract.

2. An introduction is required to provide more information about the biomedical properties of kelulut honey.

3. On what basis was the dosage fixed? Why only one particular concentration was chosen?

4. Why not compare it with a standard drug?

Author Response

Comments 1: Introduction is required more information about biomedical properties of kelulut honey.

Response 1: We thank the reviewer for pointing this out. We have made the necessary addition as follows: "KH is produced by two genera of stingless bees, Melipolini and Trigona [14,15]. It has shown promise to exert antioxidant, anti-inflammatory, and anti-apoptotic effects in the central nervous system (CNS), attributed to its high polyphenolic content [16,17]. Due to the mentioned attributes, KH is found to be a potent nootropic and neuroprotective agent [16,18]." This is found on page 2, line 62-66.

Comments 2: What basis was the dose fixed? Why only one particular concentration was chosen?

Response 2: We thank the reviewer for highlighting this. "The selected dose was previously optimized in our study which deduced the neuroprotective effects of KH on MetS-induced rats [16]." This is found on page 3, line 128-130.

Comments 3: The abbreviation should be explained before it is abbreviated. For example, MWM is mentioned in the abstract?

Response 3: Again we thank the reviewer for this issue. We have meticulously checked all the abbreviations, grammar, punctuation, and so on.

Comments 4: Why not compare it with a standard drug?

Response 4: We agree with the reviewer on this. However, due to several constraints such as availability and ethical issues, we're unable to make such comparisons.

Reviewer 2 Report

In the manuscript titled "Unveiling the therapeutic potential of Kelulut (Stingless bee) honey in Alzheimer's disease: findings from a rat model study”, the researchers examine the impact of Kelulut honey on rat models with Alzheimer’s disease.

This paper is a very interesting one because it deals with Alzheimer’s disease and this is one of the world health challenges that disrupt cognitive abilities and general well-being. It examines Kelulut honey as a new kind of therapeutic agent, which has been found to possess anti-disease properties and protect the nervous system against Alzheimer in rats. The research, therefore, adds value to previous works on Alzheimer’s disease by providing promising results based on different approaches such as behavioral tests, histochemistry and biochemical assays.

The main strength is the study is the novelty. The study fills an important information gap in the literature regarding the effects of Kelulut honey (KH) on neurodegenerative diseases, particularly Alzheimer's disease (AD). This novel exploration provides valuable insights into KH's potential as a therapeutic agent for AD, showcasing its disease-attenuating and neuroprotective properties. The comprehensive analysis and promising findings contribute significantly to the understanding and potential treatment of Alzheimer's disease.

The article fully meets the journal's requirements. It presents original research with clear and compelling findings, filling a crucial gap in the literature on the effects of Kelulut honey (KH) on neurodegenerative diseases, particularly Alzheimer's disease (AD).

I do not have any remarks or recommendations for the proposed manuscript. My overall opinion on the manuscript is that it is very well planned and performed. However, I have spotted a few technical mistakes regarding commas and spaces that should be corrected. Once these minor issues are addressed, the manuscript will be in perfect condition.

Based on the written above, after carefully considered the manuscript, I would like to suggest after correcting the technical errors, this manuscript to be published.

 However, I have spotted a few technical mistakes regarding commas and spaces that should be corrected. These are in all the text. 

Author Response

Comments 1: The reviewer addresses the few technical mistakes regarding commas and spaces that should be corrected.

Response 1: We thank the reviewer for the favorable comments made on our manuscript and we have made all the necessary corrections on the grammar, punctuation marks commas, and so on.